

# The prediction of collective Economic development based on the PSO-LSTM model in smart agriculture

Chunwu Zheng and Huwei Li

Henan Economy and Trade Vocational College, Zhengzhou, China

## ABSTRACT

Smart agriculture can promote the rural collective economy's resource coordination and market access through the Internet of Things and artificial intelligence technology and guarantee the collective economy's high-quality, sustainable development. The collective agricultural economy (CAE) is non-linear and uncertain due to regional weather, policy and other reasons. The traditional statistical regression model has low prediction accuracy and weak generalization ability on such issues. This article proposes a production prediction method using the particle swarm optimization-long short term memory (PSO-LSTM) model to predict CAE. Specifically, the LSTM method in the deep recurrent neural network is applied to predict the regional CAE. The PSO algorithm is utilized to optimize the model to improve global accuracy. The experimental results demonstrate that the PSO-LSTM method performs better than LSTM without parameter optimization and the traditional machine learning methods by comparing the RMSE and MAE evaluation index. This proves that the proposed model can provide detailed data references for the development of CAE.

## INTRODUCTION

Smart agriculture is a new farm management concept that promotes agriculture's sustainable development through multidimensional information services in production and business service (*Yin et al., 2021*; *Musa & Basir, 2021*). In the last decades, the use of heavy machinery and industrialization of the production chain has brought significant changes to agricultural production, which are crucial to the current agricultural development. Adopting smart agriculture at all levels and scales of agricultural output can meet the challenges of increasing demand for food production and decreasing the labor force (*Yue, Du & Zhang, 2021*). Smart agriculture can use different types of sensors to collect, send and receive data through communication networks, and then be managed and analyzed by management information systems and data analysis systems. Using artificial intelligence technology and machine equipment to make an efficient, reliable and accurate prediction of agricultural status reduces the corresponding production costs and environmental constraints, which is of great significance in modern agrarian activities. With the gradual

Corresponding author
Chunwu Zheng,
zhengchunwu@henetc.edu.cn

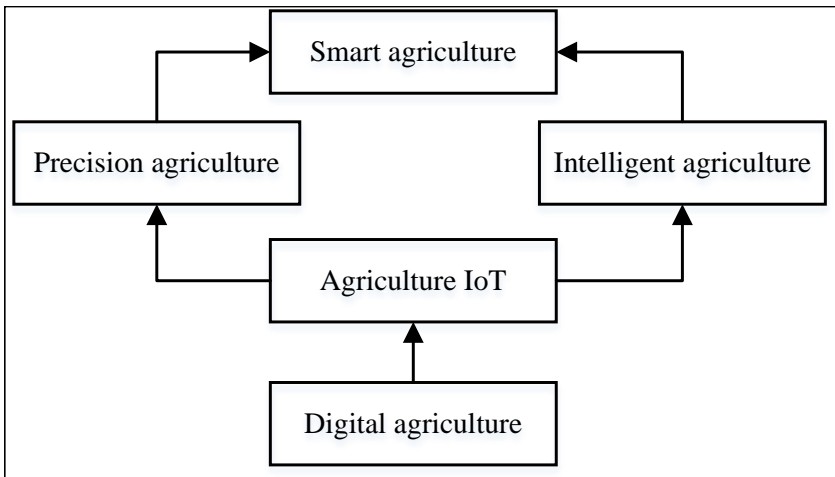

**Figure 1** **The framework of the smart agriculture.**

deterioration of the global environment, smart agriculture plays an increasingly important role (*Zhang, 2019*). The structure of intelligent agriculture is shown in Fig. 1, which includes precision agriculture, digital agriculture, agriculture IoT and precision agriculture (*Shi, Shi & Qiu, 2019*). Based on their views on industrial development, they use the Internet, IoT, cloud computing and big data to achieve intelligent control, precise investment and personalized services in agricultural production (*Ray, 2017*). The implementation subject of smart agriculture is farmers, and developing the rural collective economy is one of the fundamental driving forces to promote economic construction and social development. The comprehensive application of smart agriculture can significantly boost the development of the collective agricultural economy (CAE) in the local region by coordinating and solving the problems of limited resources and environment, poor market connectivity and low market intelligence in the rural collective economy (*Sinha & Dhanalakshmi, 2022a*). At the same time, in today's increasingly strained resources, smart agriculture can effectively coordinate all kinds of resources and facilities to realize the high modernization of agricultural production and play an essential role in promoting CAE and environmental protection. As an emerging economic model, smart agriculture is necessary for its future development to understand its development law, grasp its economic trend, and build a suitable prediction model for the development of CAE.

The development of innovative CAE involves data support from multiple industries. Therefore, economic development can be seen as a regression problem of multimodal data, while forecasting the future economy based on historical data belongs to forecasting research. No matter what kind of problem, the data itself is statistically represented as a time series. Prediction can be realized by establishing a correlation between current and past data through statistical time series analysis or machine learning methods. Time series prediction, autoregressive moving average, autoregressive summation and other methods are commonly used in statistical methods (*Rajagopalan & Santoso, 2009*; *Li et al., 2019*). When considering multiple influencing factors, generalized autoregressive

conditional heteroscedasticity (GARCH) and other models with a solid ability to describe time series volatility can be used for prediction (*Li et al., 2011*). The theory of these methods is relatively complete; Whose most significant advantage is that the model is simple. However, the corresponding accuracy is sacrificed, and the approach is relatively complex. After the model is established, a solid theoretical basis is needed for derivation and improvement. With the continuous development of computer computing capacity, methods based on artificial intelligence are emerging in prediction analysis. The main techniques include expert systems, support vector regression (SVR) and various neural network methods (*Wenxia, Xiaobo & Xi, 2014*; *Pei & Yeming, 2020*; *Bai, Xiaoning & Zhuo, 2021*). The SVR method solves the problem that selecting a neural network's architecture is challenging and sensitive to extreme value. It can also achieve satisfactory prediction results in the case of fewer samples. The neural network method has become a research hotspot in recent years. The traditional neural network retains its good friendliness with other algorithms by combining back-propagation and feedforward networks.

In contrast, the traditional neural network often ignores the time feature for data with robust time series state data. High-precision prediction results can only be achieved through more complex networks, reducing the prediction efficiency (*Huang, Wang & Che, 2020*). To better deal with time series data, the recurrent neural network (RNN) method, with the ability to analyze the potential correlation and in-depth interpretation of time series information, is skilled in time series prediction. The main application fields of RNN are natural language processing, speech recognition, time series prediction, machine translation, etc. In natural language processing, RNN can be used for text classification, emotion analysis, language model and other tasks. In the field of speech recognition, RNN can be used for audio signal processing, speech recognition, speech synthesis and other tasks. In the field of time series forecasting, RNN can be used for stock price forecasting, weather forecasting, demographic forecasting and other tasks. In machine translation, RNN can translate one language into another. The long short term memory (LSTM) method is a further step on the traditional recurrent neural network (RNN). Its unique structure design effectively avoids the phenomenon of gradient disappearance and explosion (*Zhang, Yu & Xu, 2020*). However, in the application process, neural network methods often fall into the local extremum, resulting in reduced accuracy or generalization ability.

Therefore, to more scientifically study the development trend of CAE in the context of intelligent agriculture, this article uses the historical data of CAE in this region to establish a prediction model to achieve intelligent prediction of CAE in this region. The main contributions are: A LSTM-based model is installed to deal with the CAE data to complete the economic forecast of the relevant years. The economic characteristics of different quarters are analyzed in detail, and the evaluation and analysis of the accuracy of the model when using different features are conducted. The particle swarm optimization (PSO) is employed to optimize the model to reach a higher accuracy.

The rest of this article is organized as follows: Section 2 introduces the methods used and Section 3 describes the experiment and result analysis of the agricultural economic development in this region. In Section 4, we discuss the result and the regulations that should be given to the collaborative economy. The conclusion is presented in Section 5.

## METHODS

### LSTM-based economy scale prediction

The development prediction of economic production can be seen as a multi-objective regression problem based on the current year's conditions. It can also predict future data through the relevant historical data of the economy. Such issues are prediction problems based on historical data. Therefore, a strong, robust model that can achieve multiple objectives is needed. According to the time series of historical data of multidimensional data in multi-objective regression, this article uses the RNN, which is good at processing time series, to build the model.

RNN is a classical feedback neural network, often used in regression and multivariate time series prediction analysis. Different from the back propagation (BP) neural network, RNN adds the association between information among hidden nodes to consider time information. In a typical RNN, the output of the last layer in the hidden layer is one of the inputs of the next hidden layer (*Hou, 2022*; *Nie, 2021*). Each output value of RNN is closely related to its previous state. However, RNN has only one memory unit, which leads to gradient disappearance and explosion. Insufficient memory for long sequences often occurs during complex operations. To remedy this defect, LSTM is proposed. The LSTM network model can use its unique gate structure to remember the states much better than the traditional RNN, which has greatly improved its performance in dealing with timing-related problems.

It can be found that compared with the traditional RNN structure, the LSTM cell adds a layer of cell state at the top of the cell, through which the state at all times can be transferred in the LSTM chain. The specific calculation process can be expressed by Formulas (1)–(6) (*Shi et al., 2022*):

$$f_t = \sigma(W_f \cdot [h_{t-1}, x_t] + b_f) \tag{1}$$

$$\widetilde{C}_t = \tanh(W_C \cdot [h_{t-1}, x_t] + b_C) \tag{2}$$

$$i_t = \sigma(W_i \cdot [h_{t-1}, x_t] + b_i) \tag{3}$$

$$C_t = f_t \times C_{t-1} + i_t \times \widetilde{C}_t \tag{4}$$

$$o_t = \sigma(W_o[h_{t-1}, x_t] + b_o) \tag{5}$$

$$h_t = o_t \times \tanh(C_t) \tag{6}$$

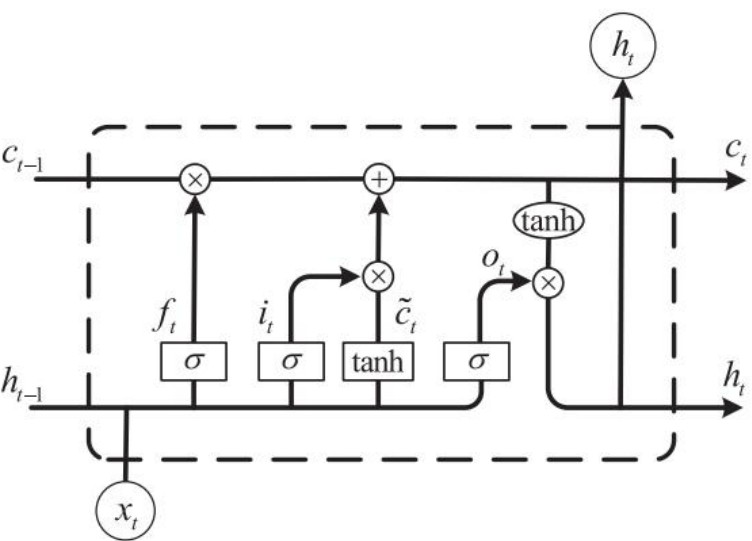

**Figure 2   The construction of the LSTM cell.**

LSTM shown in Fig. 2 can be expressed by the above formula. Compared with BP neural network and RNN, the gate structure in LSTM is the most distinctive and different from the traditional neural network. The specific implementation process is completed according to Formulas (1)–(3). First, the forgetting gate is the most crucial feature, indicating what features in Ct-1 can be used in Ct. Formula (2) $\widetilde{C}_t$ updates the cell state through the tanh function, which the input data and ht-1 can obtain. While it in Formula (3) represents the input gate, similar to ft in Formula (1), which is also calculated by the activation function $\sigma$ andht-1. After the calculation of it in Formula (3), we can determine which features can be used to update Ct. as shown in Formula (4). Finally, the output $h_t$ can be calculated through Formulas (5) and (6) to complete the next unit's input and prediction value calculation.

In Formulas (1)–(6), $W_f, W_i, W_c, W_o$ is the weight matrix corresponding to each part, $b_f, b_i, b_c^-, b_0$ is the bias. Σ and tanh both represent activation functions (*Chen et al., 2022*) which can be expressed as follows

$$\sigma(x) = \frac{1}{(1+e^{-x})} \tag{7}$$

$$\tanh(x) = \frac{(e^x - e^{-x})}{(e^x + e^{-x})}. \tag{8}$$

According to Formula (9), the final predicted value y is obtained through a complete connection layer.

$$y_t = \sigma(W_y \cdot h_t + b_y) \tag{9}$$

where, $W_y$andb_y are the weight matrix and bias, respectively.

## PSO optimization model

The neural network-related methods usually employ the gradient descent method to learn and optimize the network weight when optimizing the model, so there is a specific sensitivity to its initial value. The algorithm will converge to the local extreme if initialization is improper, significantly reducing the model accuracy. Therefore, this article plans to use the PSO method to globally optimize the initial value of LSTM to improve the network performance (*Rokbani, Abraham & Alimi, 2013*; *Kefi et al., 2016*).

The PSO initialization is to generate a swarm of particles randomly. They were then iterated to change the position of particles to obtain an optimal value. In each iteration, the particle changes its speed and position by comparing the relationship between the current one and the optimal local solution $pbes_{ti}$ and global optimal solution $gbes_{ti}$, to achieve the goal of updating the particle. When these two optimal values are obtained, we could update their states as follows:

$$
\begin{aligned}
v_i &= v_i + c_1 \times r \times (pbest_i - x_i) + \\
&\quad c_2 \times r \times (gbest_i - x_i) \\
x_i &= x_i + v_i
\end{aligned}
\tag{10}
$$

where i is the particle number in the particle swarm. r represents a random number between (0,1); $v_i$ represents the current velocity; $x_i$ is the current position; c1 and c2 is learning factor; local optimal position and global optimal position are $pbest_i$ and $gbest_i$, respectively. Parameter optimization of the LSTM model using the PSO method is shown in Fig. 3. First, the particle swarm is randomly initialized. Then, the fitness function is calculated for the model fitting effect. Then, the fitness is computed, and the best particle that meets the conditions is obtained according to the current particle state.

As shown in Fig. 3, the PSO-LSTM method used in this article first completes the application of the PSO method through parameter initialization. Then it uses its principle to update the parameters and calculate the prediction results of the LSTM. The final model can be determined after meeting the requirements of fitness, prediction accuracy, or iteration number.

## EXPERIMENT RESULTS AND ANALYSIS

To better explain the importance of CAE in economic development, this article, based on the agricultural economic output value of the region, deals with it accordingly to form a collective economy within the region to explain the prediction better. This article takes the agricultural production of the region in the past seven years, from 2015 to 2021, as an example to study the medium and long-term economic forecasting methods. The specific agricultural economic production is given in Table 1. The model is trained and verified using these data. According to the data trend, it can be seen that the agricultural economy in this region has grown year by year. We gave full play to our technological advantages through more scientific farming methods and improved the CAE.

### The result of the smart agriculture-related economy regression

The results obtained from the LSTM and PSO optimization training models introduced in Section 2.1 and 2.2 are illustrated in Fig. 4. We can find that the methods proposed in

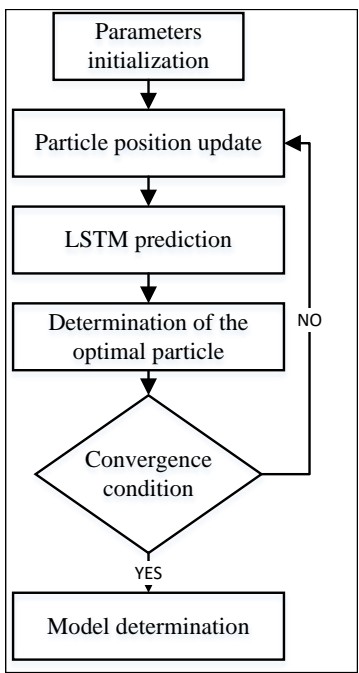

**Figure 3** The framework for the PSO-LSTM model.

**Table 1** The smart agriculture related economy.

| Year | Production/billion yuan |
| --- | --- |
| 2015 | 109.8 |
| 2016 | 118.0 |
| 2017 | 123.7 |
| 2018 | 133.8 |
| 2019 | 141.5 |
| 2020 | 148.4 |
| 2021 | 156.1 |

both trend and prediction estimation have good results. The relative errors of each year are shown in Fig. 5:

As shown in Fig. 5, the relative error of the model proposed keeps around 1%. The year with the most significant prediction error occurred in 2017, and the year with the minor error was 0.6% in 2015 and 2020. Meanwhile, to illustrate the performance of the model proposed in this article, the PSO-LSTM method used in this article is compared with the LSTM and SVR methods commonly used in machine learning. The comparison curve of the CAE production is displayed in Fig. 6.

As the production curve trend shows in Fig. 6, three models can correctly predict the economic trend, and only SVR had some deviations in 2018. To better illustrate the result of the three models, we employ Root Mean Squared Error (RMSE) and Mean Absolute

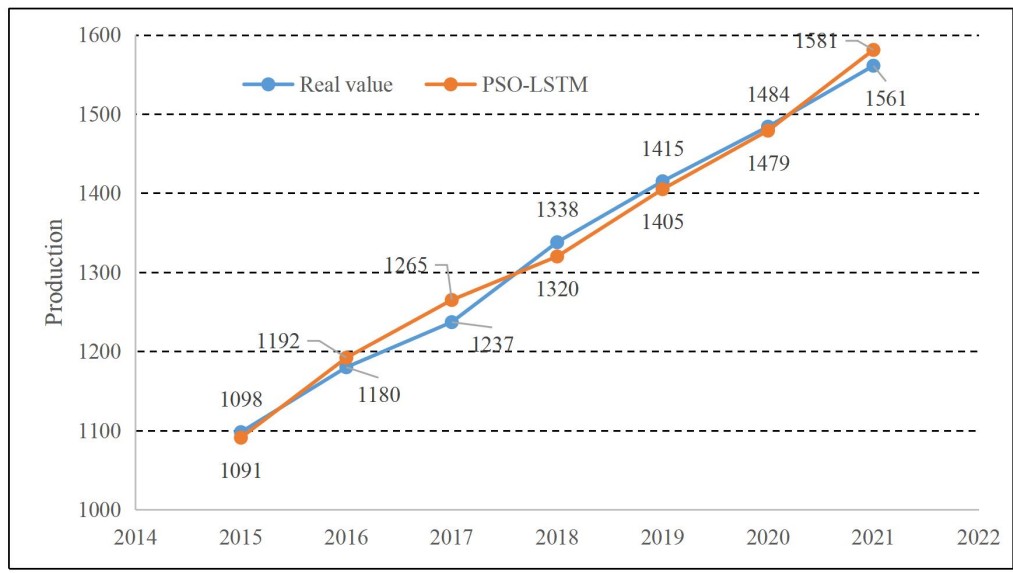

**Figure 4** Result of PSO-LSTM.

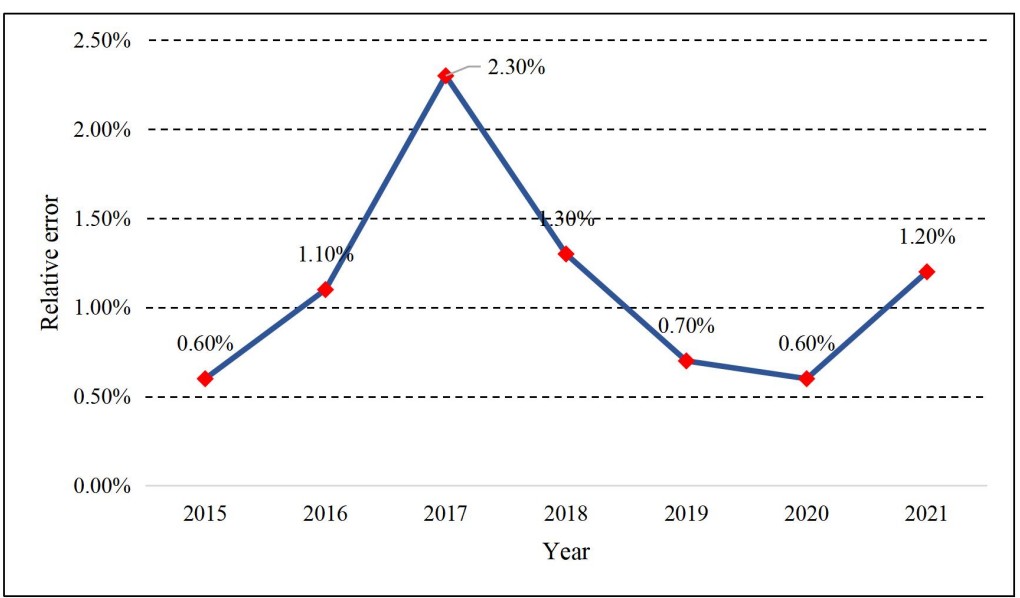

**Figure 5** Relative error of the prediction.

Deviation (MAE) to assess the model performance. The calculation methods are as follows in Formulas (11) and (12):

$$RMSE = \sqrt{\frac{1}{n}\sum_{i=1}^{n}(X_i - \hat{X})^2} \tag{11}$$

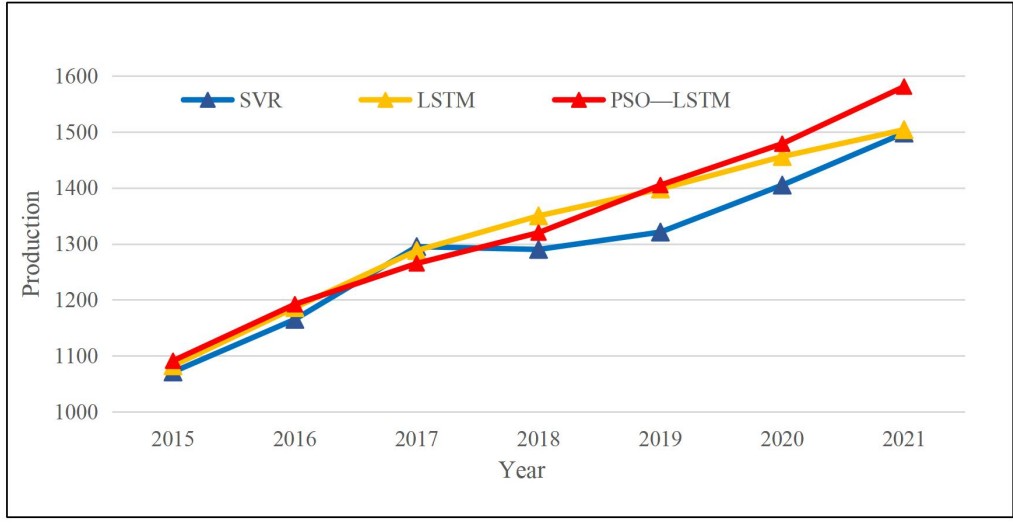

**Figure 6** **Result of different methods among different years.**

**Table 2** **The evaluation index for different methods.**

|  | RMSE | MAE |
| --- | --- | --- |
| SVR | 60.56 | 54.86 |
| LSTM | 32.42 | 26.71 |
| PSO-LSTM | **16.15** | **14.28** |

$$MAE = \frac{1}{n}\sum_{i=1}^{n}\left|X_i - \hat{X}\right| \qquad (12)$$

where X is the prediction and $\hat{X}$ is the actual value. The model accuracy will increase for these two evaluation indicators if they become smaller. The results of RMSE and MAE among the three models are given in Table 2:

As shown in Table 2, the LSTM model itself has certain advantages over the SVR method, and the advantages are further revealed after optimization through PSO.

### Prediction accuracy with historical data

To study the model in more detail, in this part, we will refine each quarter of the four years from 2018 to 2021 and forecast the data for the first quarter of this year. The accuracy rate is shown in Fig. 7:

In Fig. 7, the abscissa represents the use of different amounts of quarterly data for forecasting, and the ordinate represents the accuracy of the forecast results. Four means that only the first quarter data of each year is used for prediction. While eight means that the data from nearly eight quarters are used for prediction, and the results are significantly better than the first quarter of each year. The meanings of 12 and 16 are the same as those of eight, using the latest 12 or 16 quarters to forecast the economic output value in the first

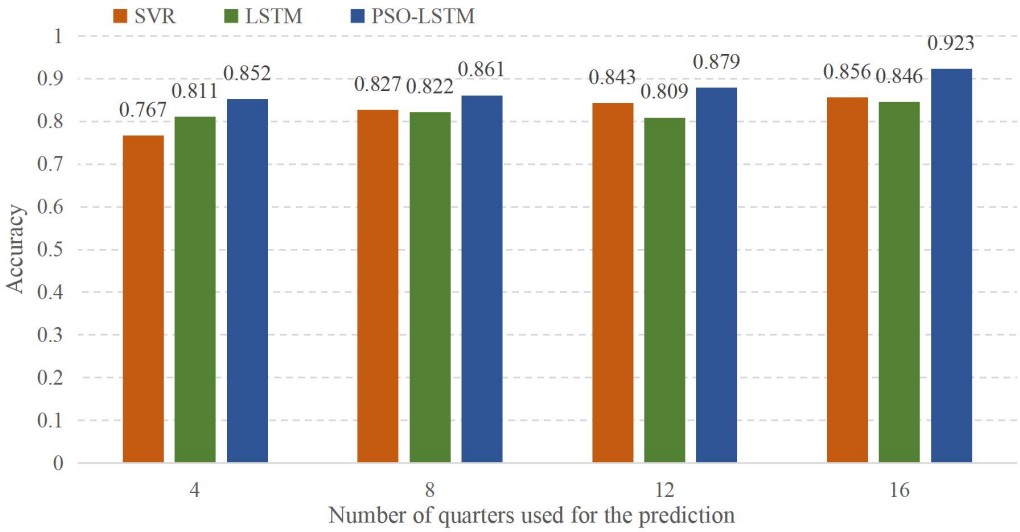

**Figure 7** Result of the current production prediction using historical data.

quarter. It can be found that the accuracy of the model prediction increases when the data volume expands. However, when using the LSTM model alone to forecast, it is found that the data for 12 quarters is not as good as the data for eight quarters. Through comparison, the model falls into the optimal local solution of the initial value, leading to a decrease in prediction accuracy.

## DISCUSSION

The sustainable development of smart agriculture requires unified allocation of resources by integrating the technology supply system, agricultural production factors and all participants in the agricultural production sector, according to the early warning and prompts issued by the smart agriculture system, and in combination with the actual local situation. Although the" IoT, including agricultural machinery, can be used to manage standard agricultural production, farmers still need to act as managers and supervisors and pay close attention to agricultural conditions (*Quy et al., 2022*). Education and training of farmers and other technical personnel should be strengthened. At the same time, it is necessary to support the use and promotion of agricultural monitoring technology and increase investment in education and training, which can promote the sustainable development of smart agriculture (*Sinha & Dhanalakshmi, 2022b*). Data transmission systems with different regulatory functions must be established to increase transparency. To ensure the continuous development of CAE, we can achieve a rapid increase in CAE by giving strong support to multiple policies and technologies. In addition to the investment in the technology and talents of smart agriculture, it is also necessary to start from CAE itself and ensure the transparency of economic data to keep its development (*Li et al., 2022*). Therefore, one of the ways to ensure data transparency is to use intelligent methods to predict the total amount of CAE and provide objective data. With the great development

of the IoT, data collection has become much more convenient so that smart agriculture will be reached with the help of AI.

In the traditional prediction analysis, the research on power prediction is the most adequate because its power use is more cyclical. As a necessity of industrial production and people's life, its fluctuation is small. Although the development of smart agriculture has made the agricultural economy grow rapidly, the research on the prediction of its economic production is insufficient. Therefore, combined with the characteristics of CAE in this region, economic growth is predicted and analyzed. Based on the historical data in recent years, the prediction model has been completed. The LSTM model optimized by PSO has a good prediction result and performs well in both RMSE and MAE evaluation indicators (*Xu et al., 2022*). Compared with the traditional RNN, the LSTM economic development prediction model with a gate structure has a better prediction and fitting effect than the traditional machine learning model. Its main advantages are: (1) LSTM has a strong learning ability and can maintain an excellent fit ability when dealing with samples of more complex time series. (2) LSTM has a unique gating structure, which can forget, retain or even update some information, effectively avoiding gradient disappearance and gradient explosion. (3) The memory unit in LSTM includes three inputs and two outputs at the current moment, which is more excellent in generalization than the traditional model (*Wang, Tong & Yu, 2020*). As a great improvement of the traditional RNN for dealing with the time series, LSTM provided a robust model for data prediction with a small amount of data. However, the neural network parameters may fall into the local extremum, so the PSO is employed to optimize the model. After the optimization of parameters, these advantages are further amplified. At the same time, using agricultural production in different quarters for analysis, we can find that as training samples expand, the prediction accuracy will gradually increase, and selecting the optimal number of prediction samples is the priority for future research.

## CONCLUSION

Aiming to predict CAE in the context of intelligent agriculture, this article proposes a CAE production prediction method using the PSO-LSTM model. The regression method has been used to fit the CAE production trend in recent years. In this article, we used the agriculture production in our region as the input, leading to output including the fitted production and predicted production among different years. The experiment result convinces us that this model has advantages over the LSTM and SVR methods with the lowest RMSE and MAE. To give more specific results, for data prediction, we refined the data in recent years and compared the prediction performance of the model with different features. The experiment using different features also proves that the method used in this article has high prediction accuracy.

However, restricted by the data scale, the available data in this article is from the last seven years, which may affect the robustness of the model proposed. As the LSTM model can handle the data, the accuracy can satisfy the demand. In the future, we will expand the data scale by using other regions and countries to improve the robustness of the model. In

the data processing, the noise filtering of time series and the intelligent selection of features, namely the required data amount, need to be further optimized.

### Funding
The authors received no funding for this work.

### Competing Interests
The authors declare there are no competing interests.

### Author Contributions
- Chunwu Zheng conceived and designed the experiments, analyzed the data, authored or reviewed drafts of the article, and approved the final draft.
- Huwei Li performed the experiments, performed the computation work, prepared figures and/or tables, and approved the final draft.

### Data Availability
The code are available in the Supplemental Files.

The data is available at Kaggle: Available at https://www.kaggle.com/datasets/syedjaferk/agriculture-commodity-data-2019.

### Supplemental Information
Supplemental information for this article can be found online at http://dx.doi.org/10.7717/peerj-cs.1304#supplemental-information.

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
