# Peer review of "The prediction of collective Economic development based on the PSO-LSTM model in smart agriculture"

_PeerJ Computer Science, doi:10.7717/peerj-cs.1304_

## Round 0.1 · original submission · Major Revisions

Please improve your article as per the suggestions of the experts and resubmit, please also improve the language and grammar professionally.

·

Basic reporting

• The title of the article is "The Prediction of Collective Economic Development based on A PSO-LSTM Model in the Context of Sustainable Development of Smart Agriculture" is too long. Please reduce it to 15 words.

Experimental design

• Please confirm whether RNN is an abbreviation, if so, please provide its full name (Line 91) when first used.

• The LSTM model is very common for economic forecasting, so it should not be the main contribution of this paper. In addition, please merge (1) and (3). (Line 101-107)

Validity of the findings

The specific calculation process expressed by Formula (1) - (6) need more citations, but I would suggest deleting them;

Additional comments

• Section 4 needs more citations, because there are a large number of concluding statements in this part, and they cannot be obtained from the experimental results of this paper.

• The conclusion is too short and needs to be supplemented with specific experimental results. Also, add the contributions of the paper at the end of the Section;

Reviewer 2 ·

Basic reporting

In this paper, the prediction model of the local smart agricultural collective economy based on the LSTM method in the deep circulation network is introduced, and the parameters of the model are globally optimized by the PSO algorithm to improve the accuracy of the model. In this paper, the model is tested based on the agricultural economic data of the local area, and the experimental results show that the proposed PSO-LSTM method has obvious advantages in RMSE and MAE evaluation indexes compared with LSTM without parameter optimization and traditional machine learning methods. This method provides detailed data reference for the development of regional collective economy under smart agriculture and has a certain promotion effect on the development of agricultural collective economy. With the current quality, this article cannot be published. This article has many defects, so my suggestion is a minor revision.
1. Please define "PSO-LSTM" in the abstract and provide its full name instead of abbreviations;

2. I think the existence of Figure 1 is not necessary, because the information conveyed by it is very limited. It is more appropriate to describe the composition structure of intelligent agriculture in words.

3 Although the background of the study was in China, the manuscript contains too many Chinese references, and I suggest replacing some of them with English references, such as Reference6 and Reference 7.

4 The title of Section 2.2 "Pso-based model optimization" seems not be appropriate, "PSO optimization" will be better;

5. Why is it that the agricultural economic data of the local area from 2015 to 2021 is taken as an example and the sample data of such a scale seems insufficient?

6. The abscissa of Figure 4 is missing;

7. The language of the conclusion needs to be further strengthened. The description of the author coincides with the abstract, and the author should focus on the outstanding contribution of the research and the application direction;

Experimental design

There is no experimental comparison of the algorithm with previously known work, so it is impossible to judge whether the algorithm is an improvement on previous work.

Validity of the findings

Please refer to basic reporting

Additional comments

Please refer to basic reporting

---

## Round 0.2 · accepted · Accept

Thank you revising the paper according to reviewer comments and good luck.

·

Basic reporting

The authors have resolved all of my raised concerns in this revision. Thus, I would like to recommend this paper for publication

Experimental design

The authors have resolved all of my raised concerns in this revision. Thus, I would like to recommend this paper for publication

Validity of the findings

The authors have resolved all of my raised concerns in this revision. Thus, I would like to recommend this paper for publication

Reviewer 2 ·

Basic reporting

I reviewed the revised version of the manuscript.
I also reviewed the rebuttal as well as the changes done in the manuscript (in highlighted form).
I noticed that authors have addressed most of the comments and have done the required changes.
I think, now manuscript is mature enough to be accepted and published.

Experimental design

I reviewed the revised version of the manuscript.
I also reviewed the rebuttal as well as the changes done in the manuscript (in highlighted form).
I noticed that authors have addressed most of the comments and have done the required changes.
I think, now manuscript is mature enough to be accepted and published.

Validity of the findings

I reviewed the revised version of the manuscript.
I also reviewed the rebuttal as well as the changes done in the manuscript (in highlighted form).
I noticed that authors have addressed most of the comments and have done the required changes.
I think, now manuscript is mature enough to be accepted and published.

Additional comments

I reviewed the revised version of the manuscript.
I also reviewed the rebuttal as well as the changes done in the manuscript (in highlighted form).
I noticed that authors have addressed most of the comments and have done the required changes.
I think, now manuscript is mature enough to be accepted and published.